# *miR-23b* and *miR-218* silencing increase Muscleblind-like expression and alleviate myotonic dystrophy phenotypes in mammalian models

Estefania Cerro-Herreros[1,2,3], Maria Sabater-Arcis[1,2,3], Juan M. Fernandez-Costa [1,2,3], Nerea Moreno[1,2,3], Manuel Perez-Alonso[1,2,3], Beatriz Llamusi[1,2,3] & Ruben Artero [1,2,3]

Functional depletion of the alternative splicing factors Muscleblind-like (MBNL 1 and 2) is at the basis of the neuromuscular disease myotonic dystrophy type 1 (DM1). We previously showed the efficacy of miRNA downregulation in *Drosophila* DM1 model. Here, we screen for miRNAs that regulate *MBNL1* and *MBNL2* in HeLa cells. We thus identify *miR-23b* and *miR-218*, and confirm that they downregulate MBNL proteins in this cell line. Antagonists of *miR-23b* and *miR-218* miRNAs enhance MBNL protein levels and rescue pathogenic missplicing events in DM1 myoblasts. Systemic delivery of these "antagomiRs" similarly boost MBNL expression and improve DM1-like phenotypes, including splicing alterations, histopathology, and myotonia in the HSA$^{LR}$ DM1 model mice. These mammalian data provide evidence for therapeutic blocking of the miRNAs that control Muscleblind-like protein expression in myotonic dystrophy.

[1] Interdisciplinary Research Structure for Biotechnology and Biomedicine (ERI BIOTECMED), University of Valencia, Dr. Moliner 50, E46100 Burjassot, Valencia, Spain. [2] Translational Genomics Group, Incliva Health Research Institute, Dr. Moliner 50, E46100 Burjassot, Valencia, Spain. [3] Joint Unit Incliva-CIPF, Dr. Moliner 50, E46100 Burjassot, Valencia, Spain. Correspondence and requests for materials should be addressed to B.L. (email: Mbeatriz.Llamusi@uv.es) or to R.A. (email: ruben.artero@uv.es)

Myotonic dystrophy type 1 (DM1) is an autosomal dominant rare genetic disease with variable presentation. It typically involves severe neuromuscular symptoms including cardiac conduction defects, myotonia, and progressive muscle weakness and wasting (atrophy). Neuropsychological dysfunction is also a common symptom of DM1[1]. The cause of DM1 is well known, namely the accumulation of mutant transcripts containing expanded CUG repeats in the 3′UTR of the *dystrophia myotonica protein kinase* (*DMPK*) gene. CUG repeats form the disease's hallmark ribonuclear foci. Mutant *DMPK* RNA triggers toxic gene misregulation events at the level of transcription[2], translation[3–6], gene silencing[7–10], alternative splicing, and polyadenylation of subsets of transcripts[11–13].

RNA toxicity stems from enhanced binding of proteins to expanded CUG RNA, which exists as imperfect hairpin structures. The RNA-binding proteins are thus depleted from their normal cellular targets. Chief among these are the Muscleblind-like proteins (MBNL1–3), whose sequestration contributes to DM1 in several ways. MBNL1 controls fetal-to-adult splicing and polyadenylation transitions in muscle and MBNL2 likely has a similar role in the brain[14,15], whereas *Mbnl3* deficit results in age-associated pathologies that are also observed in myotonic dystrophy[16,17]. No treatment has yet been specifically developed for DM1 despite intensive efforts. Numerous therapeutic approaches have been designed following different approaches[18,19] that can be broadly grouped as: (1) specific targeting of the mutant allele or its RNA product, including preventing MBNL protein sequestration using small molecules[20–23], transcriptional[24,25] and post-transcriptional silencing of *DMPK*[26], and (2) target signaling pathways downstream from CUGexp (CUG expansion) expression[27,28]. Two strategies have reached human trials: Tideglusib, a small molecule non-ATP-competitive glycogen synthase kinase 3 (GSK-3) inhibitor[4] (clinical trial NCT02858908) and IONIS-DMPKRx, an RNase H1-active ASO (antisense oligonucleotide) that target CUGexp RNA[29] (clinical trial NCT02312011). However, in IONIS-DMPKRx clinical trial, drug levels measured in muscle biopsy confirmed that the amount of target engagement was not enough to exert a desired therapeutic effect[30].

There is ample evidence that MBNL1 and MBNL2 functions are the limiting factors in DM1. Therefore, boosting their expression is a potential therapeutic avenue. Indeed overexpression of MBNL1 could rescue disease-associated RNA missplicing and muscle myotonia in a DM1 mouse model that expresses 250 CTG repeat units from a human skeletal actin promoter (HSA^LR)[31,32]. Consistently, compound loss of Muscleblind-like function reproduces cardinal features of DM1 such as reduced lifespan, heart conduction block, severe myotonia, and progressive muscle weakness[33]. *MBNL1* overexpression was well-tolerated in skeletal muscle and early and long-term *MBNL1* overexpression prevented CUG-induced myotonia, myopathy, and alternative splicing abnormalities in DM1 mice[34]. Targeted expression of *MBNL1* can even rescue eye and muscle atrophy phenotypes in *Drosophila* DM1 models[35–37].

We recently used a *Drosophila* DM1 model to show that Muscleblind could be upregulated by sequestration of repressive miRNAs to improve splicing, muscle integrity, locomotion, flight, and lifespan[38]. Here, we extend these studies to mammalian disease models and demonstrate that *miR-23b* and *miR-218* are endogenous translational repressors of *MBNL1/2* and *MBNL2*, respectively. AntagomiRs transfection upregulates MBNL proteins and rescues alternative splicing in normal and DM1 human myoblasts. Furthermore, systemic administration of antagomiRs in the HSA^LR mouse model upregulate Muscleblind-like protein in both gastrocnemius and quadriceps muscles and rescue the molecular, cellular, and functional defects of DM1 muscle.

## Results

### Identification of miRNAs that regulate *MBNL1* and *MBNL2*.
We approached a detailed description of *MBNL1* and *MBNL2* regulation by overexpressing miRNAs in HeLa cells using a commercial kit. The study identified 19 and 9 miRNAs that reduced *MBNL1* or *MBNL2* transcript levels by at least 4-fold, respectively, compared to controls (Supplementary Fig. 1). We ranked the miRNAs according to likelihood of a direct physical interaction with *MBNL1* or *MBNL2* 3′-UTR sequences (Supplementary Table 1). We selected five miRNAs with the best target predictions and also included *miR-146b* in our validation work because it downregulated *MBNL1* the most. Overall, selected miRNAs were: *miR-96* and *miR-181c* as candidate direct repressors of *MBNL1*; *miR-218* and *miR-372* as candidate repressors of *MBNL2*, and *miR-146b* and *miR-23b* as potential regulators of both.

In validation experiments, HeLa cells were transfected with the corresponding miRNA precursor sequences cloned into the *pCMV-MIR-GFP* vector. All candidate miRNAs confirmed the expected reduction in endogenous *MBNL1* and/or *MBNL2* mRNA levels (Fig. 1a, b), except for *miR-146b* that only significantly reduced *MBNL2* expression. Next, we used western blot quantification to confirm the Muscleblind-like protein downregulation by miRNAs (Fig. 1c–f). All mRNA reductions were thus confirmed at the protein level except for *miR-181c* on MBNL1 translation and *miR-146b*, which failed to repress both MBNL1 and MBNL2. Taken together, these results identified *miR-96*, *miR-23b*, and *miR-218* as new miRNAs that repress *MBNL1* and/or *MBNL2* expression both at the mRNA stability and protein levels.

### Mapping of miRNA–mRNA binding sites in the 3′UTR of *MBNL1/2*.
miRNAs generally act as post-transcriptional repressors by recognizing specific sequences in the 3′-UTR of target mRNAs[39]. To test if our three candidate miRNAs directly bind to the trailer regions of *MBNL1* and *MBNL2*, as predicted by miRanda[40] and TargetScan[41] (Fig. 2a, e), we performed reporter assays in HeLa cells. The 3′-UTR of each gene was fused to the Gaussian luciferase reporter (Gluc) so that any functional interaction of a regulatory miRNA and the reporter construct will reduce luminescence measurements. Cotransfection of miRNA target gene luciferase reporter constructs and miRNA plasmids in HeLa cells confirmed a significant decrease in the luciferase luminescence for all tested miRNAs (Fig. 2b, f).

miRNA–mRNA interaction strongly depends on the perfect complementarity between the target mRNA and the miRNA seed region at positions 2–8[39]. Next, we investigated direct binding of miRNAs to single or multiple *MBNL* 3′-UTR sequences. We made a series of miRNA target gene luciferase reporter plasmids with natural (WT), perfectly matched (PM), or absent (MUT) miRNA recognition sites. In these assays, WT versions of the 3′-UTR of *MBNL1* and *MBNL2* significantly reduced expression of the Gluc reporter when co-transfected with *miR-23b* or *miR-96* (*MBNL1*; Fig. 2c, d), and *miR-23b* or *miR-218* (*MBNL2*; Fig. 2g, h). In comparison, cotransfection with the corresponding mutant versions always abrogated the repressive effect of the miRNAs whereas PM versions had lower luciferase than WT constructs. Strikingly, deletion of any of the three *miR-218* recognition sites in *MBNL2* alleviated full repression of the Gluc reporter. Overall, we conclude that *miR-96*, *miR-218*, and *miR-23b*, directly regulate *MBNL1*, *MBNL2*, or both genes, respectively.

### *miR-23b* and *miR-218* antagomiRs stabilize MBNL transcripts.
DM1 pathology depends on Muscleblind-like protein expression in muscle. A pre-requisite for the hypothesis that miRNAs fine-

tune MBNL1 and/or MBNL2 translation in muscle was that our candidate miRNAs were expressed there. We therefore measured levels of *miR-96*, *miR-23b*, and *miR-218* in muscle by qPCR. *miR-23b* and *miR-218* were highly expressed in cultured human DM1 myoblasts and muscle biopsies, whereas *miR-96* was found at negligible levels (Fig. 3a). To test this potential of *miR-23b* and *miR-218* as therapeutic targets for DM1, we designed antisense oligonucleotides (antagomiRs) that recognize these miRNAs. AntagomiRs are very stable and have cholesterol moieties to enhance their uptake into cells[42,43]. First, we characterized their toxicity profiles. The TC10 concentrations (at which at least 90% of cells remained viable) were 654.7 nM for antagomiR-23b and 347 nM for antagomiR-218 (Fig. 3b).

Visual confirmation of cell uptake was obtained with Cy3-labeled versions of antagomiRs at concentrations ranging 50–100 nM, but not at 10 nM (Supplementary Fig. 2). Having established the effective and non-toxic range for the antagomiRs, we tested the ability of antagomiR-23b and -218 to block their corresponding miRNA at concentrations at which cell uptake was confirmed (>50 nM), but well below TC10. DM1 myoblasts were transfected with antagonists and the mRNA levels of *MBNL1* and *MBNL2* were measured 48 and 96 h later. *MBNL1* and *MBNL2* transcripts

in cells treated with antagomiR-218 increased in a dose-dependent manner at both time points and reached approximately 50% higher levels than in scramble control-treated DM1 cells (Fig. 3c, d). Of note, since silencing of *miR-218* enhanced *MBNL1* transcripts but did not bind to the *MBNL1* 3′-UTR reporter constructs (Supplementary Fig. 3) regulation by this miRNA might be indirect or dependent on sequences outside the 3′-UTR. In sharp contrast to *miR-218* blockers, the lower the concentration of antagomiR-23b the higher the increase in MBNL levels. *MBNL1* transcripts doubled in DM1 cells 96 h post-transfection with 50 nM of antagomiR-23b. Assuming a typical bell-shaped dose–response, these results suggested that working concentration for antagomiR-23b is 50 nM, or lower, and 200 nM or higher for antagomiR-218 (Fig. 3c, d). Importantly, levels of *miR-218* were not altered in cells treated with antagomiR-23b, suggesting a specific effect of the antagomiR on its target. *miR-23* family includes *miR-23a* and *miR-23b*, which are transcribed from different chromosomes, have identical seed sequences, and differ by only one nucleotide on their 3′ ends. As expected, antagomiR-23b also reduced the levels of *miRNA-23a* in the cells (Supplementary Fig. 4a, b).

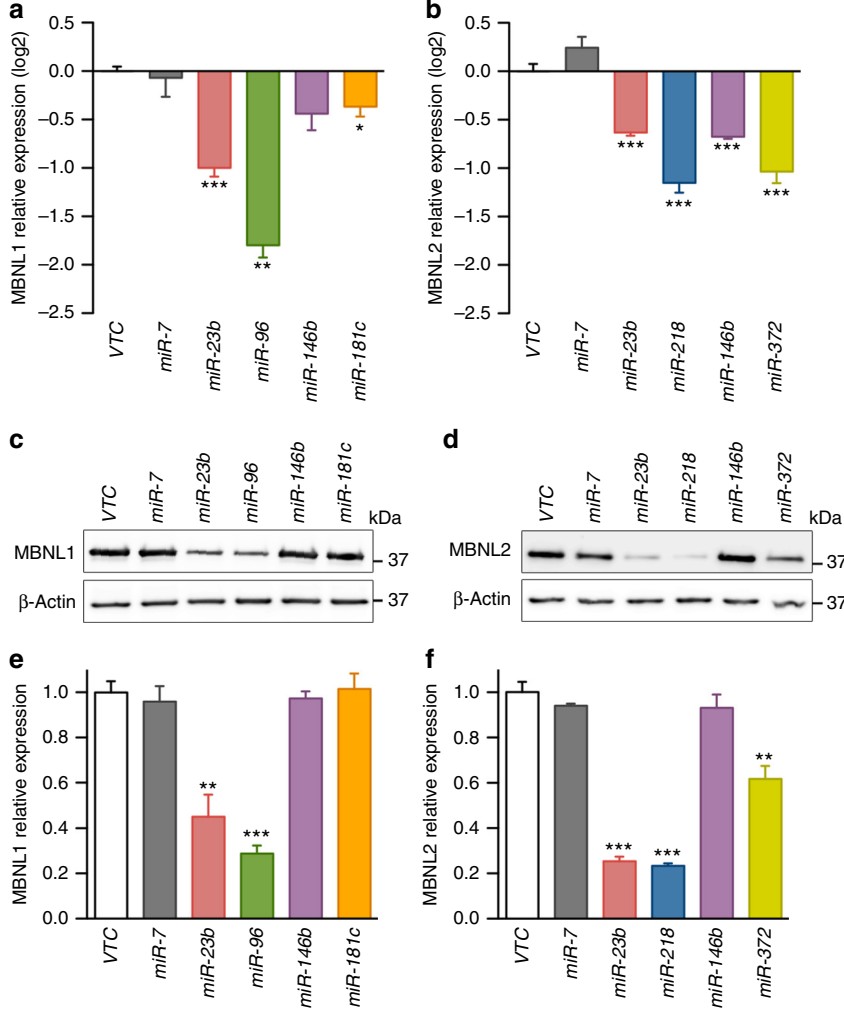

**Fig. 1** Validation of candidate *MBNL1* and/or *MBNL2* regulatory miRNAs. Logarithmic representation on base 2 (log2) of the qRT-PCR quantification of *MBNL1* (**a**) and *MBNL2* (**b**) expression relative to *GAPDH* gene in HeLa cells transfected with the indicated *pCMV-MIR* plasmids. **c**–**f** Relative protein expression levels of MBNL1 (**c**, **e**) and MBNL2 (**d**, **f**) in HeLa cells transfected as above. β-ACTIN was the endogenous control. In all cases, empty *pCMV-MIR-GFP* plasmid (VTC) was used as reference value for relative quantification, *miR-372*[66] and *miR-7* were used as positive and negative controls, respectively. GFP was used as transfection control. (*n* = 3). Data are mean ± SEM. *\*p* < 0.05, \*\**p* < 0.01, \*\*\**p* < 0.001 according to Student's *t* test

In summary, we have confirmed the ability of *miR-23b* and *miR-218* antagonists to enter DM1 cells and enhance *MBNL1* and *MBNL2* mRNA levels, at concentrations well below their toxicity threshold.

**miR-23b and miR-218 silencing rescues defects in DM1 cells.** The best-known molecular alteration in DM1 is missplicing of defined subsets of muscle transcripts. To test whether higher amounts of *MBNL1* and *MBNL2* mRNA translate into rescue of the Muscleblind-dependent splicing events, we transfected DM1 cells using the optimal conditions determined above and verified improvement of missplicing events including *Bridging integrator 1* (*BIN1*)[44], *ATPase sarcoplasmic/endoplasmic reticulum Ca²⁺ transporting 1* (*ATP2A1*)[45], *Insulin receptor* (*INSR*)[46,47], and *Piruvate kinase M* (*PKM*)[48]. Exon inclusion (Percentage Spliced In; PSI) was significantly rescued for all four transcripts 96 h post-transfection (Fig. 3e; Supplementary Fig. 5) upon *miR-23b* or *miR-218* silencing (Fig. 3f, h). Similarly, *BIN1*, *ATP2A1*, and *PKM* splicing, but not *INSR*, was also rescued 48 h after antagomiR transfection (Supplementary Figs. 5 and 6). In

contrast, increased levels of *MBNL1* and *MBNL2* in DM1 myoblasts did not significantly change the aberrant inclusion of exon 5 in the *Cardiac troponin T* (*cTNT*)[46] transcripts under any of the experimental conditions tested.

To test the specificity of antagomiRs-23b and -218, we quantified the inclusion of exon 8 of *CAPZB*, which depends on CUGBP Elav-like protein family member 1 (CELF1)[49], and observed that it was not rescued by the antagomiRs (Fig. 3e; Supplementary Figs. 5 and 6). Additionally, the regulated inclusion of exon 19 of *DLG1*, which is known to be *MBNL1* and *CELF1*-independent, did not change under any of the experimental conditions, thus discarding global effects on alternative splicing control upon antagomiR treatment (Fig. 3e; Supplementary Figs. 5, 6). Taken together, these results confirm Muscleblind-specific rescue of alternative splicing defects taking place in DM1 myoblasts as a result of antagomiR-mediated *MBNL1* and *MBNL2* derepression.

**AntagomiRs-23b/-218 restore normal MBNL cell distribution.** Since miRNAs can regulate gene expression at the mRNA

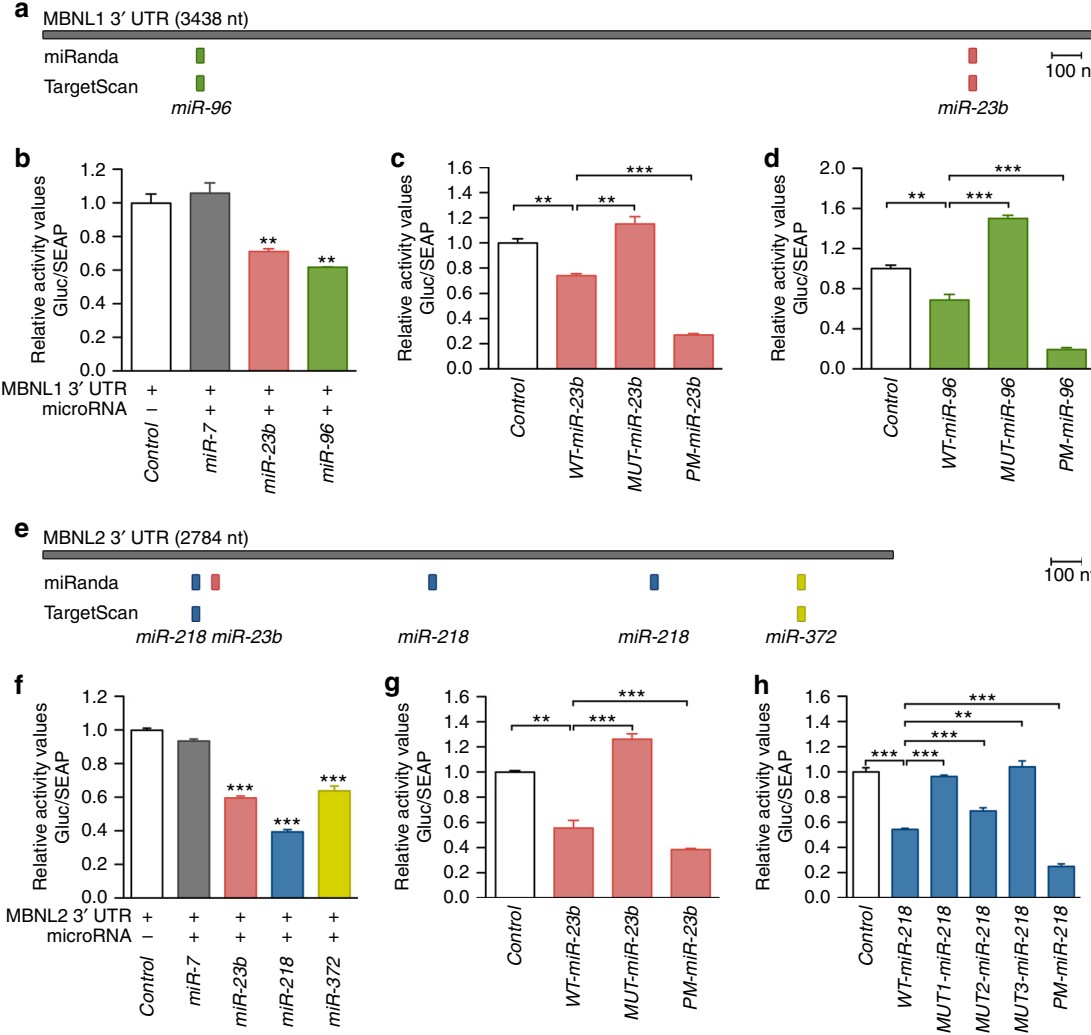

**Fig. 2** Confirmation of miRNA binding to *MBNL1* and *MBNL2* 3'UTRs. **a**, **e** Scale representation of *MBNL1* (**a**) and *MBNL2* (**e**) 3' UTRs and predicted miRNA binding sites according to miRanda and TargetScan algorithms. *MBNL1* (**b–d**) and *MBNL2* (**f–h**) 3' UTR luciferase reporter assays of HeLa cells co-transfected with wild-type (**b**, **f**) or mutated (**c**, **d**, **g**, **h**) versions of 3' UTR fused to Gaussia luciferase and miRNA plasmids ($n = 4$). *miR-7* was used as a negative control. Wild-type (WT) reporter plasmids had the natural sequence of the miRNA binding sites, mutated (MUT) constructs lacked a candidate miRNA seed region recognition site and the perfect match (PM) versions had the miRNA binding site replaced by the full complementary sequence. **p < 0.01, ***p < 0.001 according to Student's *t* test. Data are mean ± SEM

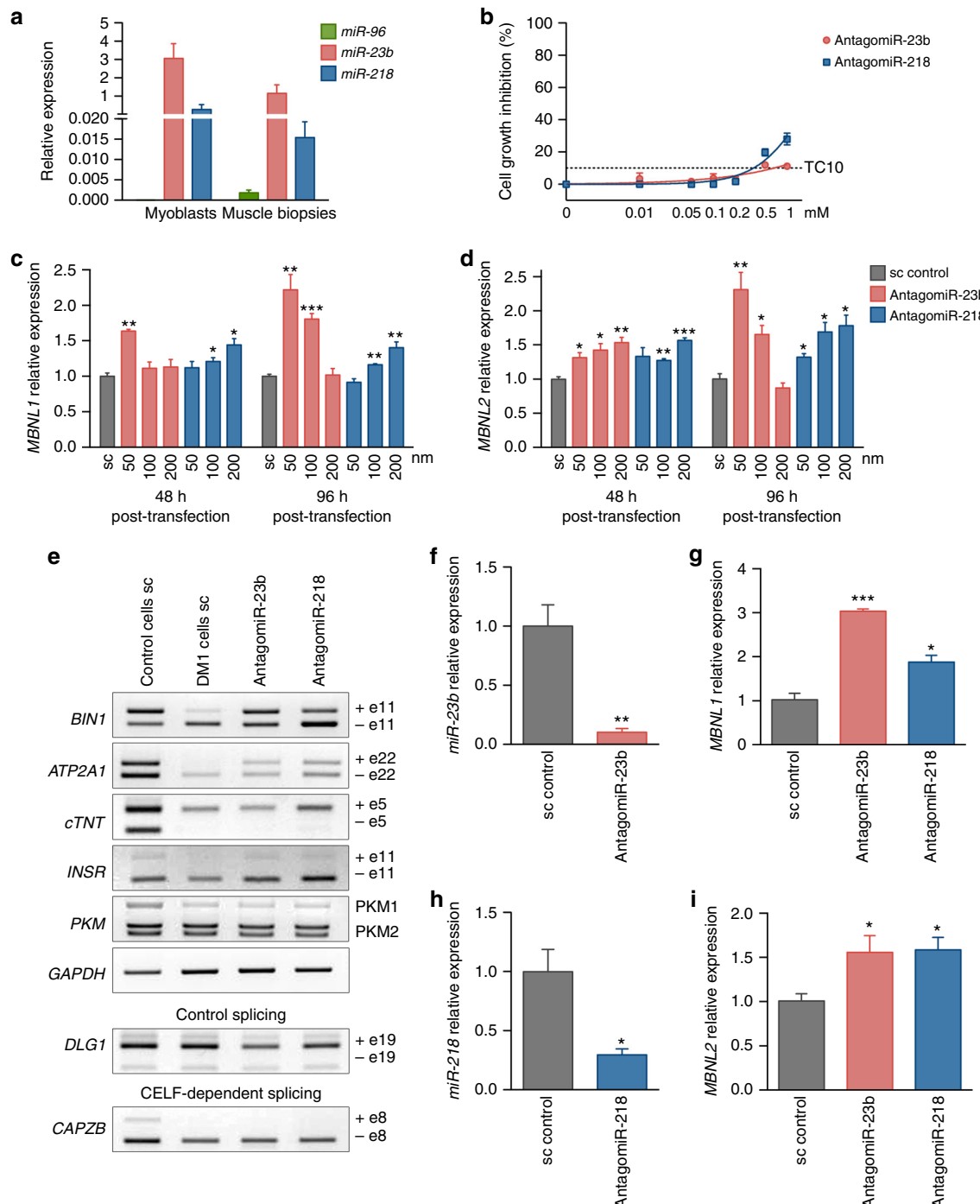

**Fig. 3** AntagomiR-23b and -218 stabilize *MBNL* transcripts and rescue alternative splicing defects in DM1 myoblasts. **a** Real-time PCR quantification of *miR-96*, *miR-23b*, and *miR-218* expression in myoblasts and muscle biopsies from DM1 patients. *U1* and *U6* snRNAs were used as reference genes. **b** Cell growth inhibition assay by MTS method. Human normal myoblasts were transfected with increasing concentrations of antagomiRs against *miR-23b* and *miR-218* (*n* = 4). TC10 was obtained using the least squares non-linear regression model. **c, d** qRT-PCR quantification of *MBNL1* (**c**) and *MBNL2* (**d**) expression relative to *GAPDH* and *ACTB* genes in human DM1 myoblasts transfected with the indicated antagomiRs or scrambled control antagomiR (sc). **e** Semiquantitative RT-PCR analyses of splicing events altered in *BIN1* (exon 11), *ATP2A1* (exon 22), *cTNT* (exon 5), *INR* (exon 11), and *PKM* isoforms in DM1 cells. *GAPDH* was used as internal control. Inclusion of *DLG1* exon 9, which is not altered in DM1, and *CAPZB* exon 8, which is CELF1-dependent, were used as additional controls. **f, h** miRNA real-time PCR determination of available *miR-23b* (**f**) or *miR-218* (**h**) in DM1 myoblasts 96 h after transfection with 50 nM of antagomiR-23b or 200 nM of antagomiR-218. *U1* and *U6* snRNAs were used as reference genes in **f** and **h**. **g, i** qRT-PCR analyses of *MBNL1* (**g**) and *MBNL2* (**i**) expression relative to *GAPDH* and *ACTB* genes in human myoblasts 96 h after transfection with 50 nM of antagomiR-23b or 200 nM of antagomiR-218. (*n* = 3). Data are mean ± SEM. *$p < 0.05$, **$p < 0.01$, ***$p < 0.001$ according to Student's *t* test

stability and translation levels, we sought to determine the effect of antagomiRs on MBNL1 and MBNL2 protein expression. Upon antagomiR treatment, qPCR data confirmed a significant increase in the levels of *MBNL1* and *MBNL2* mRNA 96 h (Fig. 3g, i) or 48 h (Supplementary Fig. 6) post-transfection. At the protein level, these differences were further enhanced and western blots detected 4–5-fold more MBNL1, and 3–5-fold higher MBNL2 proteins, in DM1 myoblasts after 96 h (Fig. 4a, b, d, e) and 48 h (Supplementary Fig. 6) of antagomiR treatment. Of note, MBNL1 and MBNL2 protein levels remained unaltered in control and DM1 cells mock-transfected or transfected with a scrambled control antagomiR (Supplementary Fig. 7). In contrast, CELF1 protein levels remained unchanged upon *miR-23b* or *miR-218* silencing (Fig. 4c, f; Supplementary Fig. 6) and, consistently, *CAPZB* alternative splicing remained the same. Importantly, this increase was clearly visible by immunofluorescence. Whereas both MBNL1 and MBNL2 were sequestered in ribonuclear foci of DM1 myoblasts (Fig. 4h, l), antagomiRs-23b and -218 robustly increased the protein expression and restored their distribution in the cytoplasm and in the cell nucleus (Fig. 4i, j, m, n). The increase of MBNL1 and MBNL2 proteins in the cell nucleus was consistent with the previously shown splicing rescue. Because the relationship between MBNL proteins and CUGexp foci formation is complex, a potential undesirable side effect of boosting MBNL expression was an increase in the number of ribonuclear foci. To specifically test this hypothesis, we quantified foci in antagomiR-treated DM1 fibroblasts and found that remained unaltered (Fig. 4o–r).

***miR-23b* and *miR-218* are expressed in tissues relevant to DM1**. To obtain a broader view of the expression pattern and relative expression of these miRNAs, we turned to mouse tissue samples, using the reference FVB strain. We analyzed muscle (quadriceps, gastrocnemius), whole heart, and central nervous system (forebrain, cerebellum, hippocampus). All samples had robust expression of *miR-23b* and *miR-218*, whereas *miR-96* was again almost undetectable except in cerebellum (Fig. 5a). *miR-218* was expressed up to 80 times higher in brain tissues than in muscle-derived samples. *miR-23b* and *miR-218* are therefore strongly expressed in the tissues that are highly relevant to DM1 pathology (skeletal muscle, heart, brain) where they likely repress MBNL1 and/or MBNL2 translation. Since levels of MBNL proteins are limiting in DM1, *miR-23b* and *miR-218* may be contributing to the disease phenotypes and can be regarded as potential therapeutic targets.

**AntagomiRs increase Mbnl expression in HSA<sup>LR</sup> mouse muscle**. Next, we investigated the activity of antagomiR-23b and -218 in the HSA<sup>LR</sup> mouse DM1 model[31]. First, we evaluated the ability of antagomiRs to reach skeletal muscle. Cy3-labeled versions of the antagomiRs were administered to a 4-month-old HSA<sup>LR</sup> mouse by a single subcutaneous injection. Four days post-injection, hind limb gastrocnemius and quadriceps muscles were processed to detect the labeled oligonucleotide. We observed the antagomiRs by anti-Cy3 immunofluorescence in a strong punctate pattern in nuclei and membrane of both muscles. The antagomiRs were also diffusely present throughout the cells (Fig. 5b–e). These data demonstrate that antagomiR oligonucleotides that block *miR-23b* or *miR-218* can reach the skeletal muscles of a DM1 mouse model.

We decided to inject unlabeled antagomiRs to nine additional DM1 animals, in three consecutive injections (12 h intervals) to a final dose of 12.5 mg kg$^{-1}$. The controls were injected with PBS1x ($n = 10$) or scrambled antagomiR ($n = 5$). Four days after the first injection, animals were sacrificed and quadriceps and gastrocnemius were obtained for histological and molecular analysis. We confirmed that *miR-23b* and *miR-218* were strongly silenced by their complementary antagomiRs. *miR-23b* was reduced to 30–40% and miR-218 to 50% of the levels measured in the untreated HSA<sup>LR</sup> mice (Fig. 5f, g). As a result of miRNAs reduction, *Mbnl1* and *Mbnl2* increased at the transcript and protein levels in both muscle types (Fig. 5h, i, j, k, m, n). Importantly, levels of Celf1 protein were not altered by either treatment (Fig. 5l, o). In mice injected with PBS or with scrambled oligo as control, target microRNAs and the *Mbnl1* and *Mbnl2* transcript levels were indistinguishable (Supplementary Fig. 8a–d).

**AntagomiRs rescue missplicing of muscle transcripts in mice**. Given the robust increase in Mbnl1 and 2 in treated gastrocnemius and quadriceps muscles, we sought to confirm a rescue of Mbnl-dependent splicing events *Atp2a1*, *Chloride channel protein 1 (Clcn1)*, and Nuclear factor 1 X-type (*Nfix*) in HSA<sup>LR</sup> mice (Fig. 6a, b; Supplementary Fig. 9). AntagomiR administration ameliorated aberrant exon choices for *Atp2a1* (exon 22) and *Nfix* (exon 7), and increased *Clcn1* exon 7a PSI in gastrocnemius but not in quadriceps of HSA<sup>LR</sup> mice. To test the specificity of antagomiRs-23b and -218 on MBNL regulation, we quantified the inclusion of exon 8 of *Capzb*, Exon 21 of *Ank2*, and exon 3 of *Mfn2*, which depend on Celf1[49,50], and observed that they were not altered by treatment with antagomiRs.

In a routine test of transgene expression of HSA<sup>LR</sup> mice, we discovered that CUG expression levels varied up to 0.5-fold among animals and that variation positively correlates with aberrant inclusion of alternative exons in gastrocnemius and quadriceps (Supplementary Fig. 10). To note *Atp2a1* exon 22 inclusion was bimodal. Two out of 10 mice that expressed low levels of transgene included exon 22 to levels significantly higher (closer to normal) than the rest of HSA<sup>LR</sup> mice and were therefore excluded from the analysis (Supplementary Figs. 10 and 11). These data suggest that the lower the expression of CUG repeat RNA in muscles the less missplicing there is. In contrast, in the antagomiR-treated HSA<sup>LR</sup> muscle samples splicing defects correlated with *Mbnl* mRNA levels, instead of repeat expression, which supported a causal role of these proteins in the rescue of the splicing events (Supplementary Fig. 10). Despite the intrinsic variability of the model, we conclude that both antagomiRs achieved similar levels of rescue in all gastrocnemius-missplicing events. However, antagomiR-23b rescued *Nfix* splicing to a greater extent than antagomiR-218 in quadriceps, which correlated with the lower upregulation of Mbnl1 and 2 protein levels achieved by antagomiR-218 in this muscle. Consistent with the unchanged levels of Celf1 protein in the muscles of treated HSA<sup>LR</sup> mice, inclusion percentage of *Celf1*-dependent splicing events in gastrocnemius and quadriceps of treated and control mice was very similar (Fig. 6a, b and Supplementary Fig. 9). Importantly, the Mbnl1 and 2 protein levels and the splicing patterns of Mbnl- or Celf1-dependent events in mice injected with PBS or with scrambled oligo, as control, were indistinguishable (Supplementary Fig. 8e–i).

These results indicate that systemic delivery of antagomiRs was able to rescue muscle missplicing in vivo in a DM1 mouse model.

**AntagomiRs improve muscle histopathology and reduce myotonia**. Defective transitions of fetal to adult alternative splicing patterns have been proposed to originate DM1 muscle phenotypes[51]. In HSA<sup>LR</sup> DM1 model mice, alterations in ionic currents cause repetitive action potentials, or myotonia, that can be quantified by electromyography. Before treatment, all DM1 mice had grade 3 or 4 myotonia, i.e., abundant repetitive

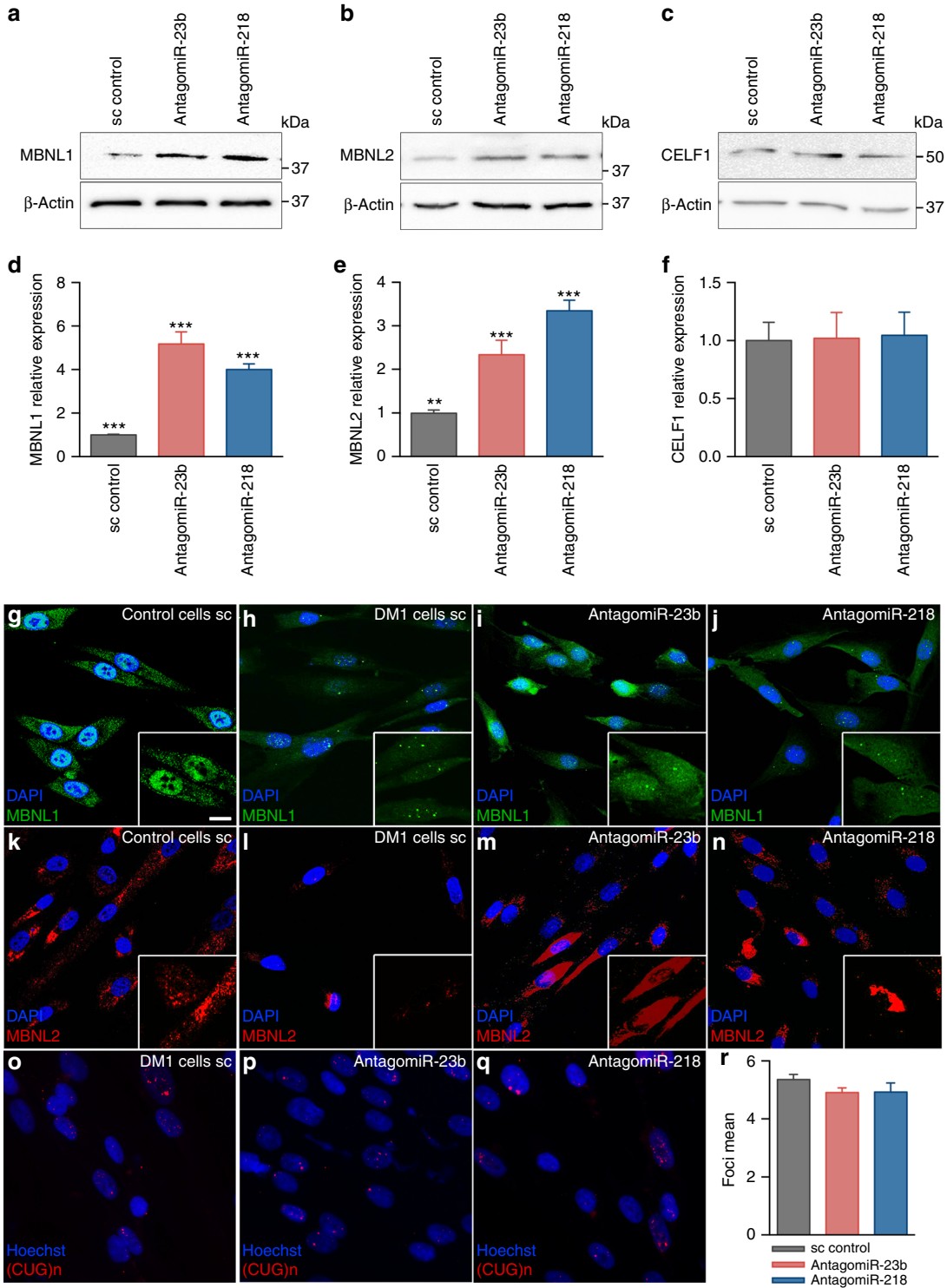

**Fig. 4** Increase of MBNL1 and MBNL2 upon silencing of *miR-23b* or *miR-218* in human myoblast. **a–f** Western blot quantification of MBNL1 (**a**, **d**), MBNL2 (**b**, **e**), and CELF1 (**c**, **f**) expression levels in DM1 human myoblast 96 h after transfection with 50 nM of antagomiR-23b, 200 nM of antagomiR-218 or a scrambled control antagomiR (sc). β-ACTIN expression was used as endogenous control ($n = 3$). Data are mean ± SEM. **p < 0.01, ***p < 0.001 in Student's *t* test. **g–n** Representative confocal images of MBNL1 (green) and MBNL2 (red) staining in healthy controls (control cells) and DM1 human myoblast 96 h after transfection with antagomiRs against *miR-23b* (50 nM) or *miR-218* (200 nM) and a scrambled control antagomiR (DM1 cells). Nuclei were counterstained with DAPI (blue). In DM1 cells, endogenous MBNL1 (**h**) and MBNL2 (**l**) were in nuclear aggregates (green and red puncta) and the total amount of both was reduced compared to control cells (**g**) and (**k**), respectively. In contrast, DM1 cells treated with antagomiRs against *miR-23b* or *miR-218* showed a robust increase in cytoplasmic and nuclear MBNL1 (**i**, **j**) and MBNL2 (**m**, **n**) levels compared to DM1 cells. **b–q** Representative fluorescence of FISH images showing (CUG)n RNA foci (red) in DM1 human fibroblasts transfected with antagomiRs against *miR-23b* (50 nM) or *miR-218* (200 nM) and a scrambled control antagomiR. Nuclei were counterstained with Hoechst (blue). AntagomiRs did not significant change the number of ribonuclear foci in DM1 fibroblasts (**r**). Scale bar = 20 μm

discharges with the vast majority of electrode insertions. Four days after, antagomiRs reduced myotonia to grade 2 (myotonic discharge in >50% of insertions) or grade 1 (occasional myotonic discharge) in 55% of the mice treated with antagomiR-218, and in 50% of the mice treated with antagomiR-23b, respectively (Fig. 6c).

A typical histological hallmark of DM1 and HSA$^{LR}$ mouse muscle fibers is a central location of nuclei, which results from myopathic muscle attempting to regenerate[5]. Both antagomiRs caused decentralization of nuclei in both gastrocnemius and quadriceps muscles (Fig. 6d–h). In contrast, myotonia levels and number of central nuclei remained unaltered in mice treated with the scrambled antagomiR (Supplementary Fig. 8j–m).

Taken together, these results validate the potential of antagomiR-23b and -218 as a drug that suppresses CUG-repeat RNA-induced myopathy in mammals.

**AntagomiR long-term treatment rescue functional phenotypes.** In order to assess the long-term effects of the antagomiR treatments, we studied characteristic molecular and functional alterations in the HSA$^{LR}$ mice in groups of mice treated with the same dose and posology as before but sacrificed 6 weeks after the antagomiR injections. The levels of *miR-23b* and *miR-218* in these mice were still significantly reduced 6 weeks after the initiation of the experiment, but the reduction was less pronounced than in the short-term treatment. In general, the reduction of target miRNAs was not enough to maintain the increased Mbnl transcript levels, as only Mbnl1 transcripts in quadriceps were significantly augmented 6 weeks after injection (Fig. 7a–d). Myotonia grade measured before injection (bi), in the halfway point (hp, 3 weeks after the injection) and in the final point (fp, before sacrifice) showed a clear tendency to decrease with time (Fig. 7e), and forelimb muscle force measured in the final point and normalized to body weight, increased in both treatments

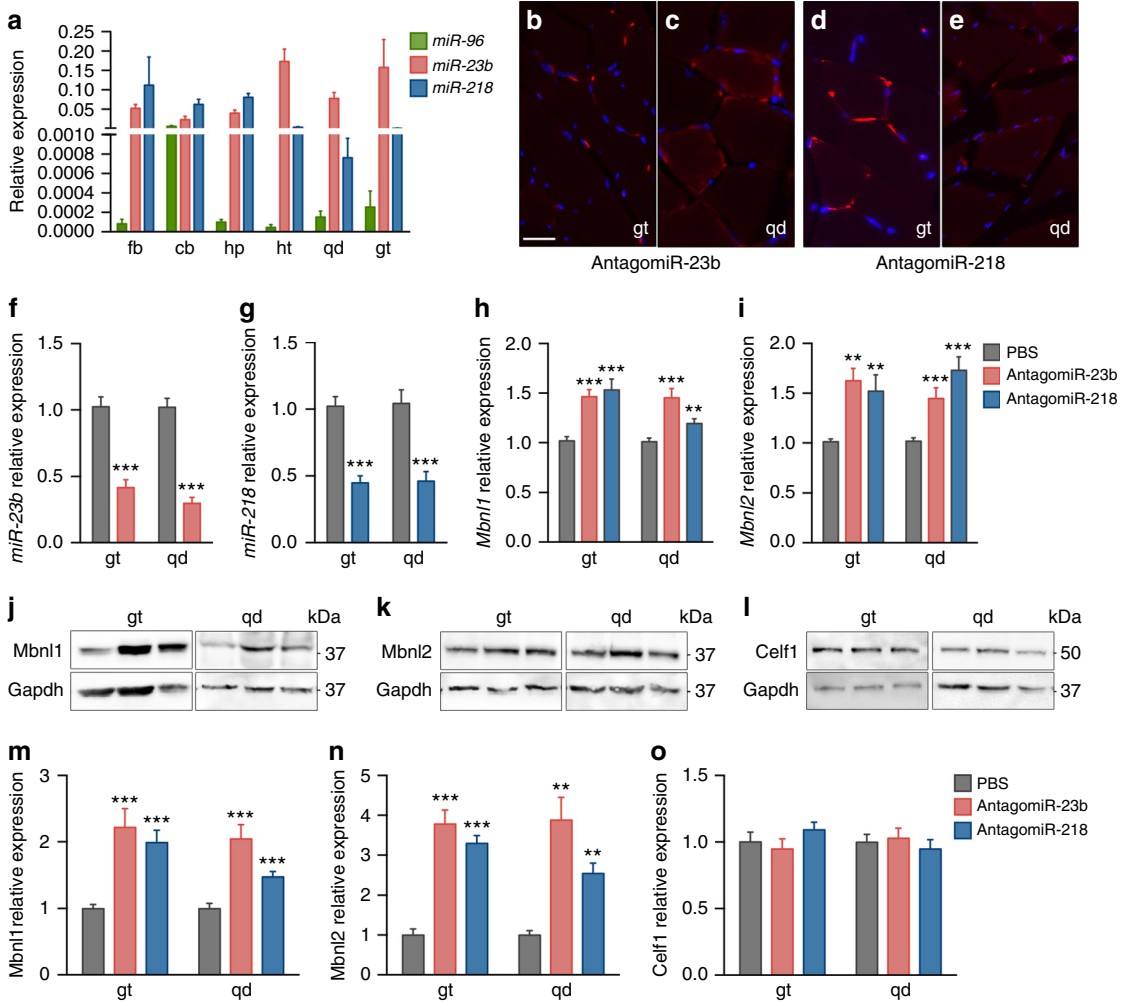

**Fig. 5** Subcutaneous injection of antagomiR-23b or antagomiR-218 in HSA$^{LR}$ mice reduced target miRNA levels and increased Mbnl1 and Mbnl2 without affecting levels of Celf1. **a** qPCR quantification of *miR-96*, *miR-23b*, and *miR-218* expression levels in forebrain (fb), cerebellum (cb), hippocampus (hp), heart (ht), and quadriceps (qd) and gastrocnemius (gt) muscles ($n = 3$). Average expression levels of *U1* and *U6* were used for normalization. **b–e** Immunodetection of Cy3-labeled antagomiRs in gastrocnemius (**b**, **d**) and quadriceps (**c**, **e**) cryosections of HSA$^{LR}$ treated mice ($n = 1$). Myonuclei were counterstained with DAPI (blue). Scale bar = 50 μm. **f–g** Quantification of *miR-23b* and *miR-218* in gt and qd muscles of untreated mice (PBS, gray bars) or treated with antagomiR-23b (pink bars) or antagomiR-218 (blue bars). Relative values (to average of *U1* and *U6* expression) were further normalized to the levels in untreated mice. **h, i** Real-time PCR quantification of *Mbnl1* and *Mbnl2* transcript levels in gt and qd muscles. Expression levels relative to the endogenous *Gapdh* were normalized to the levels in untreated mice. **j–o** Western blotting analysis of Mbnl1 (**j**, **m**), Mbnl2 (**k**, **n**), and Celf1 (**l**, **o**) proteins in mouse gt and qd muscles. Representative blots used for quantification in (**m–o**) are shown in (**j–l**). The data were analyzed by unpaired Student's *t* test compared to untreated HSA$^{LR}$ mice. Data are mean ± SEM. *$p < 0.05$, **$p < 0.01$, ***$p < 0.001$; HSA$^{LR}$ PBS ($n = 10$ in **f–o**), HSA$^{LR}$ antagomiR-23b ($n = 9$, in **f–o**). HSA$^{LR}$ antagomiR-218 ($n = 9$, in **f–o**)

although the difference was only significant for antagomiR-218 (Fig. 7f).

Importantly, visual necropsy and biochemical blood parameters measured of the mice before sacrifice support a non-deleterious effect of the treatment with the antagomiRs, even after 6 weeks (Supplementary Data 2). The only parameters altered in these analyses were total bilirubin, which was decreased in all treated animals, potentially reflecting an increase in liver metabolic rate, and monocyte number, which increased also in all treated animals, suggesting activation of the immune system. Given that these two parameters were also altered in the scrambled-treated mice, the alterations might be caused by the oligo chemistry instead of specific microRNA inhibition.

## Discussion

A largely unexplored therapeutic strategy for DM1 is therapeutic gene modulation (TGM), which pursues to raise or lower the expression of a given gene to alleviate a pathological condition. Examples of TGM are inhibition of CD44 in metastatic prostate cancer[52], or the pharmacological enhancement of *utrophin* expression to compensate lack of dystrophin in Duchenne Muscular Dystrophy[53]. Previous attempts to raise the critically low levels of Muscleblind in DM1 involved epigenetic upregulation of endogenous *MBNL*[54] or derepression of *muscleblind* by sponge-mediated silencing of miRNAs in a *Drosophila* DM1 model[38]. Oligonucleotide-based modulation of miRNA activity has

prompted great attention because of its efficacy in animal models of disease and the development of specialized chemistries[52,55,56].

In this study, we identified *miR-23b* and *miR-218* as inhibitors of *MBNL1* and *MBNL2* translation, and show that complementary antagomiRs robustly silence their target miRNAs in patient-derived myoblasts and HSA^LR mouse model. We found that antagomiRs are effective at doses lower than those previously reported in muscle[43,57,58]. By inhibiting *miR-23b* and *miR-218* in mouse muscle, we were able to upregulate MBNL1 and MBNL2 protein levels by approximately over 2-fold and 4-fold, respectively, without affecting CELF1 levels. Importantly, MBNL protein overexpression was previously shown to be well-tolerated in mouse models[34]. Accordingly, in our study, upregulation of MBNL proteins through *miR-23b* or *miR-218* silencing was not harmful, as mice showed no detrimental phenotype 6 weeks after the treatment. Similarly, *miR-23b* silencing in the heart of an inducible DM1 mice model caused no overt phenotype, despite producing CELF1 overexpression[59]. This data supports the safety of our therapeutic approach in DM. Although different effects of antagomiR-23b on CELF1 levels are reported in these two studies, antagomiR dose used or tissue-specific effects might explain this controversy. Given the tissue-specific expression of microRNAs, their targeting in TGM is also advantageous because it regulates genes only in the places where the regulator miRNAs are expressed, and the intensity of the upregulation will depend on the degree of repression by that microRNA.

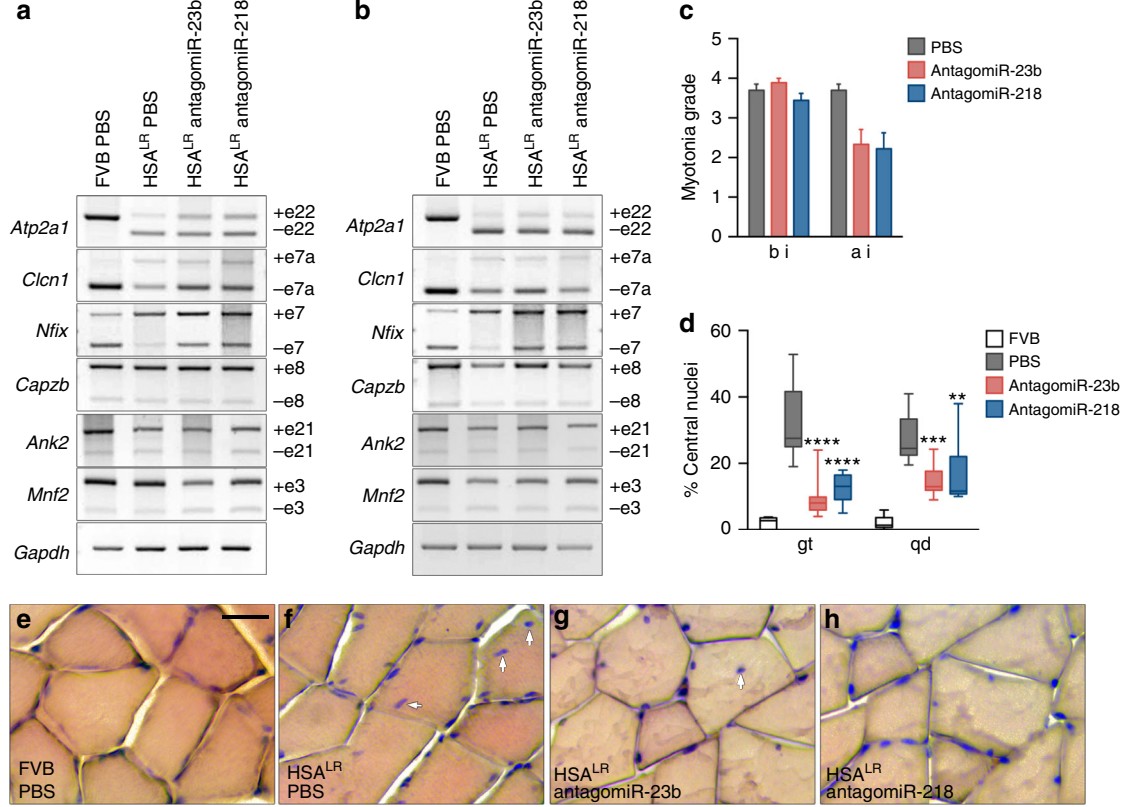

**Fig. 6** Systemic delivery of antagomiRs improved missplicing of Mbnl-dependent transcripts, myotonia, and muscle histopathology in HSA^LR mice. **a**, **b** RT-PCR analyses of the splicing of *Atp2a1* exon 22, *Clcn1* exon 7a, *Nfix* exon 7, *Capzb* exon 8, *Ank2* exon 21, and *Mnf2* exon 3 in gastrocnemius (gt) (**a**) and quadriceps (qd) (**b**) muscles. *Gadph* values were used for normalization in the quantification of the exon inclusion in Supplementary Figs. 9 and 11. **c** Electromyographic myotonia grade in antagomiR (pink and blue bars) or PBS-treated (gray bars) HSA^LR mice before (bi) and 4 days after injection (ai). Data are mean ± SEM. **d** Quantification of the percentage of muscle fibers with central nuclei in gt and qd muscles of control FVB (white bar), and PBS (gray bar) or antagomiR-treated (pink and blue bars) HSA^LR mice. **e**–**h** Representative hematoxylin and eosin staining of gt muscles from all four groups of mice. Arrows point to examples of centrally located nuclei in muscle fibers. Scale bar = 50 μm. The data were analyzed by unpaired Student's *t* test compared to untreated HSA^LR mice. Data are media ± SEM. *$p < 0.05$; FVB ($n = 3$), HSA^LR PBS ($n = 5$), HSA^LR antagomiR-23b ($n = 4$) and HSA^LR antagomiR-218 ($n = 4$)

Several previous studies have reported that phenotypes of HSA[LR] model mice are intrinsically variable[4,60]. This was suggested to stem from somatic instability of CUG repeats[4], but we found that the levels of expression of the HSA[LR] transgene and the magnitude of missplicings have a strong positive correlation. Indeed, for some splice events such as *Atp2a1*, small changes in transgene expression translated into big changes in exon inclusion. We suggest that the quantification of the transgene expression is an important step in order to identify outliers.

It is well known that sequestration of MBNLs and activation of CELF1 block the developmental change from fetal to adult RNA transcripts in DM1 muscle. We demonstrate that upregulated MBNL1 and MBNL2 was sufficient to rescue several Muscleblind-dependent, but not the CELF1-dependent splice events, in DM1 myoblasts and HSA[LR] DM1 model mice. Interestingly, a mere 2-fold increase of Mbnl2 (and a marginal increase in Mbnl1) protein in quadriceps was sufficient to strongly rescue *Nfix* missplicing. Thus, we found that a relatively small upregulation of the MBNL proteins can have a profound impact on DM1 phenotypes not only at the level of missplicing but also at histopathology (central nuclei) and functional (myotonia and muscle strength) levels. Overall our study highlights the use of oligonucleotide drugs to specifically de-repress the expression of the *MBNL1* and *MBNL2* genes as therapeutic approach for DM1.

## Methods

**Cell culture.** HeLa cells were obtained from Dr. Francisco Palau (Sant Joan de Deu Hospital, Spain) and were grown in Dulbecco's Modified Eagle's Medium (DMEM) with 1 g L$^{-1}$ glucose, 1% penicillin and streptomycin (P/S), and 10% fetal bovine serum (FBS; Sigma). Unaffected (control) and patient-derived cells (DM1 cells carrying 1300 CTG repeats quantified in the blood cells)[61] were kindly provided by Dr. Furling (Institute of Myology, Paris). Fibroblast cells were grown in DMEM with 4.5 g L$^{-1}$ glucose, 1% P/S, and 10% FBS (Sigma). To transdifferentiate fibroblasts into myoblasts by inducing MyoD expression, the cells were plated in muscle differentiation medium (MDM) containing DMEM with 4.5 g L$^{-1}$ glucose, 1% P/S, 2% horse serum, 1% apo-transferrin (10 mg ml$^{-1}$), 0.1% insulin (10 mg ml$^{-1}$), and 0.02% doxycycline (10 mg ml$^{-1}$). In all cases, the cells were grown at 37 °C in a humidified atmosphere containing 5% CO$_2$.

**MicroRNA profiling.** To search for miRNAs that regulate *MBNL1* and *MBNL2* in HeLa cells, the SureFIND Cancer miRNA Transcriptome PCR Array (Qiagen) was used according to the manufacturer's instructions. Briefly, a multiplex quantitative real-time PCR (qRT-PCR) assay was set up using the QuantiFast Probe PCR Kit reagent with TaqMan probes gene expression assays for human *MBNL1* and *MBNL2* (FAM-labeled probe) and *GAPDH* (MAX-labeled probe; Qiagen). qRT-PCR was performed using a StepOnePlus real-time thermal cycler and the results were analyzed using Excel-based data-analysis software provided with the SureFIND miRNA Transcriptome PCR Array. *MBNL1* and *MBNL2* gene expression was normalized to GAPDH.

**Computational prediction of miRNA targets in MBNL1/2 3′ UTRs.** Information about the predicted miRNA binding to *MBNL1* and *MBNL2* was obtained from miRecords[62] and miRDIP[63] databases. These databases integrate nine different programs (microT, MiRanda, MirTar2_V4.0, Mir Target2, Pic Tar, PITA, RNA hybrid, RNA 22, and TargetScan) that predict miRNA targets. A miRNA was considered as a candidate regulator if it was predicted by at least eight of these prediction algorithms. TargetScan (release 6.2) and MiRanda (release August 2010) were used to design the 3′UTR reporter assay, which allows analysis of the miRNA–mRNA binding sites.

**AntagomiRs.** Cy3-labeled and non-labeled oligonucleotides were synthesized by Creative Biogene. The antagomiR sequences were as follows:

5′-mG*mG*mUmAmAmUmCmCmCmUmGmGmCmAmAmUmGmU*mG*mA*mU*-3′-chol (antagomiR-23b)

5′-mA*mC*mAmUmGmGmUmAmGmAmUmCmAmCmGmCmA*mC*mA*mA*-3′-chol (antagomiR-218)

5′-mC*mA*mGmUmAmCmUmUmUmUmGmUmGmUmA*mC*mA*mA*-3′-chol (scramble control, SC)

where m denotes 2′-*O*-methyl-modified phosphoramidites, * denotes phosphorothioate linkages, and chol denotes cholesterol groups. Cy3-labeled oligonucleotides were used to visualize the distribution of the compound in cells and mouse tissues.

**Cell transfection.** HeLa cells and human myoblasts were transfected using X-tremeGENE™ HP (Roche) according to the manufacturer's protocols. For the miRNA overexpression assay, HeLa cells were seeded in 6-well plates at approximately 80% confluence and transfected with 1 μg of miRNA precursor sequences cloned into the *pCMV-MIR-GFP* vector (OriGene). At 48 h post-transfection, the cells were harvested for the qRT-PCR and western blot analyses. For the reporter assays, the cells were seeded in 24-well plates at approximately 80% confluence and transfected with MBNL1 and MBNL2 3′UTR luciferase constructs cloned into a *pEZX-MT05-Gluc* vector (GeneCopoeia). For each transfection, 500 ng of the appropriate reporter construct and 500 ng of the appropriate miRNA plasmids

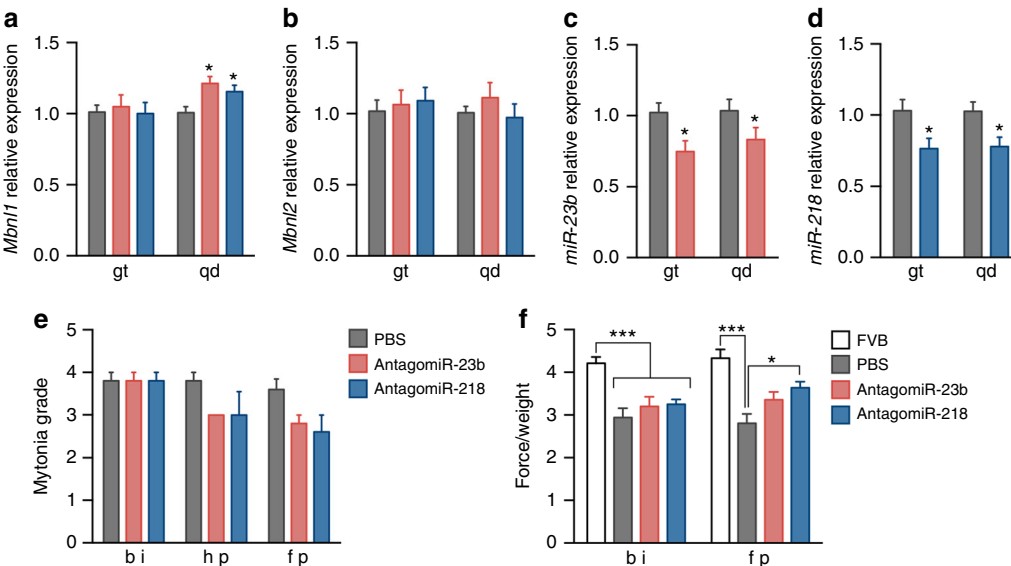

**Fig. 7** Muscle function is improved in HSA[LR] mice 6 weeks after injection of antagomiRs. **a**, **b** qRT-PCR quantification of *Mbnl1* and *Mbnl2* transcript levels in gt and qd muscles of untreated mice (PBS, gray bars) or treated with antagomiR-23b (pink bars) or antagomiR-218 (blue bars). **c**, **d** Real-time PCR quantification of *miR-23b* and *miR-218*. **e** Electromyographic myotonia grade in antagomiR (pink and blue bars) or PBS-treated (gray bars) HSA[LR] mice before (bi), in the halfway-point (hp) and 6 weeks after injection (final point, fp). **f** Forelimb grip strength of mice treated with antagomiRs measured before injection (bi) and 6 weeks after injection (fp). The grip force was normalized with the body weight of each mouse. The data were analyzed by unpaired Student's *t* test compared to HSA[LR] mice treated with the vehicle PBS. Data are mean ± SEM. *$p < 0.05$, ***$p < 0.001$; FVB ($n = 5$), HSA[LR] PBS ($n = 6$), HSA[LR] antagomiR-23b ($n = 5$) and HSA[LR] antagomiR-218 ($n = 5$)

(OriGene) were used. In the case of *miR-23b*, *miR-96*, and *miR-218*, three types of constructs were tested, wild-type (WT), perfect match (PM), and constructs with a deletion in the seed region (MUT) predicted to disrupt binding. Both PM and MUT were provided by GeneCopoeia (see Supplementary Data 1). The supernatant was collected 48 h after transfection and the reporter activity was assayed.

Control fibroblasts were seeded into 96-well plates ($1 \times 10^5$ cells per well), transfected with different antagomiR concentrations (from 1 nM to 1 μM) and transdifferentiated into myoblasts for 96 h in MDM medium; to analyze the toxicity. As a control to test the transfection levels, the Cy3-antagomiR was transfected in separate experiments under the same conditions. For the RT-PCR, qRT-PCR, and Western blot assays, control and DM1 fibroblasts were plated into petri dishes ($1 \times 10^6$ cells per well), transfected with the appropriate antagomiR, and then differentiated for 48 or 96 h. Additionally, for the RT-PCR, control fibroblasts were plated into petri dishes ($1 \times 10^6$ cells per well), transfected with the scrambled control antagomiR, and then differentiated for 48 or 96 h. Finally, for the immunofluorescence and foci assay, fibroblasts were seeded into 24-well plates ($3 \times 10^5$ cells per well), transfected with the relevant antagomiRs, and transdifferentiated into myoblasts for 96 h.

**Transgenic mice and antagomiR administration**. Mouse handling and experimental procedures conformed to the European law regarding laboratory animal care and experimentation (2003/65/CE) and were approved by Conselleria de Agricultura, Generalitat Valenciana (Uso de bloqueadores de miRNAS como terapia potencial en distrofia miotónica, reference number 2016/VSC/PEA/00155). Homozygous transgenic HSA$^{LR}$ (line 20 b) mice[31] were provided by Prof. C. Thornton (University of Rochester Medical Center, Rochester, NY, USA) and mice with the corresponding genetic background (FVB) were used as controls. Age-matched HSA$^{LR}$ (<5 months old) male mice received three subcutaneous injections of 100 μl of 1× PBS (vehicle) ($n = 10$) or antagomiR (antagomiR-23b $n = 9$, antagomiR-218 $n = 9$, and antagomiR-SC $n = 5$) delivered to the interscapular area every 12 h. The overall quantity of antagomiR finally administered divided among the three injections was 12.5 mg kg$^{-1}$. For the long-term treatment, the same injection procedure was used in mice of 3.5 months in age (PBS, antagomiR-23b, antagomiR-218 and antagomiR-SC $n = 5$). Four days and 6 weeks after the first injection, the mice were sacrificed and the tissues of interest were harvested and divided into two samples each. One part was frozen in liquid nitrogen for the molecular analyses, and the other was fixed in 4% paraformaldehyde (PFA) and cryoprotected in 30% sucrose before histological processing. Cy3-labeled antagomiRs were administered in a single subcutaneous injection of 10 mg kg$^{-1}$.

**RNA extraction, RT-PCR, and qRT-PCR**. Total RNA from HeLa cells, human myoblasts, and murine muscle tissues was isolated using the miRNeasy Mini Kit (Qiagen; Valencia, CA) according to the manufacturer's instructions. One microgram of RNA was digested with DNase I (Invitrogen) and reverse-transcribed with SuperScript II (Invitrogen) using random hexanucleotides; 20 ng of cDNA was used in a standard PCR reaction with GoTaq polymerase (Promega). Specific primers were used to analyze the alternative splicing of *BIN1*, *ATP2A1*, *INR*, *PKM*, *cTNT*, *CAPZB*, and *DLG1* in control and DM1 human myoblasts, and *Atp2a1*, *Clcn1*, *Nfix*, *Capzb*[54], *Ank2, and Mfn2* in mouse samples (quadriceps and gastrocnemius). *GAPDH* and *Gapdh* were used as endogenous controls using 0.2 ng of cDNA. In the case of *PKM*, PCR products were digested with *Pst*I (Thermo Scientific™). PCR products were separated on a 2% agarose gel and quantified using ImageJ software (NIH). The primer sequences and exons analyzed are provided in Supplementary Table 2.

We used 1 ng of HeLa, human myoblast, or mouse tissue cDNA as a template for multiplex qRT-PCR using the QuantiFast Probe PCR Kit reagent. Commercial TaqMan probes (Qiagen) were used to detect human (*MBNL1* and *MBNL2*) or mouse (*Mbnl1* and *Mbnl2*; FAM-labeled probes) and reference (*GAPDH*; MAX-labeled probes) genes. Results from myoblasts were normalized to *GAPDH* and *ACTB* (TAMRA-labeled probe; Integrated DNA Technologies) whereas the mouse results were normalized to *Gapdh* only. HSA transgene expression levels were determined by qRT-PCR as described previously[64].

miRNA expression in human DM1 myoblasts, muscle biopsies, and murine tissues (muscle, heart, and central nervous system) was quantified using specific miRCURY™-locked nucleic acid microRNA PCR primers (Exiqon) according to the manufacturer's instructions. Relative gene expression was normalized to *U1* or *U6* snRNA.

Expression levels were measured using an Applied Biosystems StepOnePlus Real Time PCR System. Expression relative to the endogenous gene and control group was calculated using the $2^{-\Delta\Delta Ct}$ method. Pairs of samples were compared using two-tailed *t* tests ($\alpha = 0.05$), applying Welch's correction when necessary. The statistical differences were estimated by the Student's *t* tests ($p < 0.05$) on normalized data. Uncropped agarose gels are shown in Supplementary Figures 13 and 16.

**Western blotting**. For total protein extraction, HeLa and human myoblast cells were sonicated while mouse muscles (gastrocnemius and quadriceps) were homogenized in RIPA buffer (150 mM NaCl, 1.0% IGEPAL, 0.5% sodium deoxycholate, 0.1% SDS, 50 mM Tris–HCl, pH 8.0) supplemented with protease and phosphatase inhibitor cocktails (Roche Applied Science). Total proteins were quantified with a BCA protein assay kit (Pierce) using bovine serum albumin as a standard concentration range. For the immunodetection assay, 20 μg of samples were denatured for 5 min at 100 °C, electrophoresed on 12% SDS-PAGE gels, transferred onto 0.45 μm nitrocellulose membranes (GE Healthcare), and blocked with 5% non-fat dried milk in PBS-T (8 mM Na$_2$HPO$_4$, 150 mM NaCl, 2 mM KH$_2$PO$_4$, 3 mM KCl, 0.05% Tween 20, pH 7.4).

For HeLa cells, human myoblast, and murine samples, membranes were incubated overnight at 4 °C either with primary mouse anti-MBNL1 (1:1000, ab77017, Abcam) or mouse anti-CUG-BP1 (1:200, clone 3B1, Santa Cruz) antibodies. To detect MBNL2, mouse anti-MBNL2 (1:100, clone MB2a, Developmental Studies Hybridoma Bank) was used for human myoblast and mouse samples while rabbit anti-MBNL2 (1:1000, ab105331, Abcam) antibody was used for HeLa cells. All primary antibodies were detected using horseradish peroxidase (HRP)-conjugated anti-mouse-IgG secondary antibody (1 h, 1:5000, Sigma-Aldrich), except for the MBNL2 antibody in HeLa cell samples, which required a HRP-conjugated anti-rabbit-IgG secondary antibody (1 h, 1:5000, Sigma-Aldrich).

Loading controls were the anti-β-ACTIN antibody (1 h, 1:5000, clone AC-15, Sigma-Aldrich) for cell samples and anti-Gapdh (1 h, 1:5000, clone G-9, Santa Cruz) for mouse samples, followed by HRP-conjugated anti-mouse-IgG secondary antibody (1 h, 1:5000, Sigma-Aldrich). Immunoreactive bands were detected using an enhanced chemiluminescence Western Blotting Substrate (Pierce) and images were acquired with an ImageQuant LAS 4000 (GE Healthcare). Quantification was performed using ImageJ software (NIH), and statistical differences were estimated using Student's *t* test ($p < 0.05$) on normalized data. Uncropped westerns are shown in Supplementary Figures 13, 14, and 15.

**Luciferase reporter assay**. The activity of Gaussian luciferase (GLuc) and alkaline phosphatase (SEAP) were measured by quantifying the luminescence present in conditioned medium using the secreted-pair dual luminescence kit (GeneCopoeia) according to the manufacturer's protocols. Gaussian luciferase activity was normalized to alkaline phosphatase activity (GLuc/SEAP). The values of the luciferase activity were determined using a Tecan Infinite M200 PRO plate reader (Life Sciences). Statistical differences in the data were estimated using Student's *t* test ($p < 0.05$) on normalized data.

**Cell proliferation assay**. Cells were seeded at $10^5$ cells per ml in 96-well plates and transfected with antagomiRs, as previously explained; 96 h post-transfection, cell proliferation was measured using the CellTiter 96® AQueous Non-Radioactive Cell Proliferation Assay (Promega) following the manufacturer's instructions. The TC$_{10}$ and dose–response inhibition curves were calculated using non-linear least squares regression, and absorbance levels were determined using a Tecan Infinite M200 PRO plate reader (Life Sciences).

**Immunofluorescence methods**. For MBNL1 and MBNL2, myoblasts were fixed with 4% PFA for 15 min at room temperature (RT) followed by several washes in 1× PBS. Cells were then permeabilized with PBS-T (0.3% Triton-X in PBS) and blocked (PBS-T, 0.5% BSA, 1% donkey serum) for 30 min at RT, and incubated either with primary antibody mouse anti-MBNL1 (1:200, ab77017, Abcam) or rabbit anti-MBNL2 (1:200, ab105331, Abcam) at 4 °C overnight. After several PBS-T washes, the cells were incubated for 1 h with a biotin-conjugated secondary antibody, and anti-mouse-IgG (1:200, Sigma-Aldrich) to detect anti-MBNL1 or anti-rabbit-IgG (1:200, Sigma-Aldrich) to detect anti-MBNL2. The fluorescence signal was amplified with an Elite ABC kit (VECTASTAIN) for 30 min at RT, followed by PBS-T washes and incubation with either streptavidin-FITC (1:200, Vector) to detect anti-MBNL1 or streptavidin-Texas Red (1:200, Vector) to detect anti-MBNL2, for 45 min at RT. After several washes with PBS, the cells were mounted with VECTASHIELD® mounting medium containing DAPI (Vector) to detect the nuclei.

The Cy3 moiety was synthetically attached to the 5′ end of the oligonucleotide to visualize the distribution of the compound. Frozen sections (10 μm) of mouse tissues including heart, brain, gastrocnemius, and quadriceps were immunostained using anti-Cy3 antibody (1:50, Santa Cruz) followed by a secondary goat biotin-conjugated anti-mouse-IgG (1:200, Sigma-Aldrich). Cy3-labeled antagomiRs were directly detectable under a fluorescence microscope in myoblast cells. In all cases, the nuclei were stained with DAPI. Images of myoblast cells were taken on an Olympus FluoView FV100 confocal microscope and images of human myoblast and mouse tissues containing Cy3-antagomiR were obtained using a Leica DM4000 B LED fluorescence microscope. In all cases, the images were taken at a 40× magnification and processed with Adobe Photoshop software (Adobe System Inc.).

**Fluorescent in situ hybridization**. Fibroblasts were aliquoted into 8-well Cell Culture Slide ($3 \times 10^5$ cells per well) and transfected with the antagomiRs. In situ detection was performed as previously described[65]. Images were taken and analyzed using an IN Cell Analyzer 2200 Imaging System (GE Healthcare).

**Electromyography studies**. Electromyography was performed before the treatment, at the halfway point and at the time of sacrifice under general anesthesia, as

previously described[32]. Briefly, five needle insertions were performed in each quadriceps muscle of both hind limbs, and myotonic discharges were graded on a five-point scale: 0, no myotonia; 1, occasional myotonic discharge in ≤50% of the needle insertions; 2, myotonic discharge in >50% of the insertions; 3, myotonic discharge in nearly all of the insertions; and 4, myotonic discharge in all insertions.

**Muscle histology**. Frozen 15 μm-sections of mouse gastrocnemius and quadriceps muscles were stained with haematoxylin eosin (H&E) and mounted with VEC-TASHIELD® mounting medium (Vector) according to standard procedures. Images were taken at a 100× magnification with a Leica DM2500 microscope. The percentage of fibers containing central nuclei was quantified in a total of 500 fibers in each mouse.

**Forelimb grip strength test**. The forelimb grip strength was measured with a Grip Strength Meter (BIO-GS3; Bioseb, USA). The peak pull force (measured in grams) was recorded on a digital force transducer when the mouse grasped the bar. The gauge of force transducer was reset to 0 g after each measurement. Tension was recorded by the gauge at the time the mouse released its forepaws from the bar. We performed three consecutive measurements at 30 s intervals. The bodyweight measurement was performed in parallel. The final value is obtained by dividing the average value of the grip force with the body weight of each mouse.

**Data availability**. All relevant data are available within the manuscript and its supplementary information or from the authors upon reasonable request. Please contact Rubén Artero (Ruben.artero@uv.es) for any inquire.

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

## Acknowledgments

This work was funded by a grant from the Ministerio de Economía y Competitividad (SAF2015-64500-R, including funds from the European Regional Development Fund) to R.A. E.C.H. was supported by a predoctoral fellowship (BES-2013-064522) from the Ministerio de Economia y Competitividad. J.M.F.C. was supported by a postdoctoral fellowship (APOSTD/2017/088) from the Conselleria d' Educació, Investigació, Cultura i Esport (Generalitat Valenciana). Authors also thank Andres Rey Mellado for his technical help and Inmaculada Noguera, veterinary head of the animal facilities at the University of Valencia who performed the mouse necropsies.

## Author contributions

R.A. provided the conceptual framework for the study. All the authors conceived and designed the experiments. E.C.H., M.S.A., N.M., J.M.F.C., and B.L.L. performed the experiments and analyzed data. E.C.H., J.M.F.C., and B.L.L. prepared the manuscript with input from M.P-A. and R.A.

## Additional information

**Competing interests:** The method described in this paper is the subject of a patent application (inventors: E.C.H., J.M.F.C., B.L.L., R.A.). The remaining authors declare no competing interests.

