## [Peer Review File · Nature Communications]

Reviewers' comments:

Reviewer #1 (Remarks to the Author):

This paper describes the identification of microRNAs and antagomirs to target them to affect expression of MBL1 protein in DM1., The authors have used model cellular systems to identify a series of microRNAs that regulate the expression of the protein. These targets were first validated, again, in a model cellular system. Then, the targets were measured for their expression in DM1 patient cells and also in the mouse model for DM1. The application of the antagomirs boost expression of MBNL1 protein. In addition, these studies were applied into animal models of disease. The experiments were carefully carried out.

a few comments

1. The citations of the small molecule approaches that target CUG repeats is incomplete and much of the more advanced approaches have not been cited properly. For example, a recent Nature chemical biology paper for various approaches to target should be referenced. Additionally, there are several approaches such as transcriptional inhibition, HDAC inhibition, and also RNA targeting that should be better described to provide a landscape type of description of the approaches
2. The targeted microRNAs are involved in a myriad of processes. For example, 23b is a tumor suppressor and 218 is deleted in various cancers. Thus, what is the effect long term on other proteins in these microRNA-mediated pathways and also what are the effects on these proteins. This is an important consideration that is not even described in the manuscript.
3. What are the overall effects of over expression of MBNL1? There is also a loss of DMPK and more MBNL1 could cause more of the target repeat to be retained in the nucleus. Does that mean more of the translational defect of this protein is observed, does it also mean that the RNA is more abundant because loaded with more protein. This should be described and these effects explained in the text
4. oligonucleotides-based approaches have been put into the clinical for DM1 and they have been taken out of the clinic because of a lack of muscle penetration into human muscle. This should be described and an explanation of why this approach could be better, or why likely it is not, should be vetted out in the paper
5. Other microRNAs could be targeted with the oligonucleotides at the dosage given and, yet, there are no selectivity studies done. For example, microRNAs that have overlapping seed sequence or are partially complimentary should be studied and their effects measured. This is important for a variety of reasons, and this is an additional consideration to point 2 above

Reviewer #2 (Remarks to the Author):

Cerro-Herrero s et al. shown that miR-23b and miR-218 repress MBNL1/2 and MBNL2. Using antagomiR against miR-23b and miR-218 there was an upregulation of the MBNL proteins which rescued the alternative splicing in DM1 human myoblasts. In vivo antagomir administration in the HSA mouse model upregulated Muscleblind-like protein in gastrocnemius and quadriceps muscles. This was associated with an attenuation of impaired molecular signaling and improvements in muscle function.

Questions

The control mice received an injection of PBS. A more appropriate control would be the injection of

a non-specific scramble antagomir. What is the justification for injecting only PBS and not a control antagomir?

Why would the antagomiR administration only ameliorate aberrant exon choices for Atp2a1 (exon 22) and Nfix (exon 7), and increased Clcn1 exon 7a PSI in gastrocnemius but not in quadriceps of HSALR mice. Does this suggest that such an approach would not be therapeutically relevant for humans?

Statistics

Why have the authors only used 4 mice per group? What is the justification for such a small sample size?

Page 8, line 3. "..., antagomiR-23b rescued Nfix and Clcn1 splicing to a greater extent than antagomiR-218 in quadriceps, which correlated with the lower upregulation of Mbnl1 and 2 protein levels achieved by antagomiR-218 in this muscle". How reliable is a correlation with only 4 data points?

Figure 5a. Why was only n=3 analysed for the PCR? The figure legend states that 5 and 4 mice were used.

Discussion

Page 9, second paragraph. The authors state that no detrimental phenotype was observed over the 4 day experimental period. What exactly were the authors looking for that would suggest a "detrimental phenotype"? What happens if the mice are analysed several weeks after the antagomirs are injected? This is important from a pre-clinical perspective.

Also, the authors state that "Targeting miRNAs in TGM is also advantageous because it regulates genes in specific tissues ". I fail to see how this approach is tissues specific. This becomes even more relevant as the authors show that the approach does not target all muscles. This suggests that this approach would not be suitable for humans. Please comment.

Reviewer #3 (Remarks to the Author):

Following their earlier studies in *Drosophila*, Cerro-Herreros and colleagues screened for human miRNAs that modulate MBNL1 and MBNL2 levels in HeLa cells and identify miR-96, miR-23b and miR-218 as novel miRs that decrease MBNL RNA and protein levels. Luciferase reporter assays with MBNL 3' UTRs confirmed this regulatory activity and when predicted binding sites were mutated, or made perfect matches, regulation was altered accordingly. Although miR-23b and miR-218 were expressed in myoblasts and muscle, miR-96 was not and sampling of mouse tissues confirmed that miR-23b and miR-218 were expressed at relatively high levels in the majority of tissues examined so these miRs were pursued further. In the interest of developing treatments that increase MBNL protein levels, antagonists of candidate miRNAs were designed. ASO/antagomiRs against -23b and -218 increased DM1 myoblast MBNL RNA and protein levels and rescued missplicing of some MBNL target RNAs. AntagomiR treatment of HSALR mice also increased MBNL protein levels in two skeletal muscles and correlated with improvements in myotonia and histopathology as well as partial rescue of alternative splicing. Overall, this is a novel and important study that demonstrates these miRNAs modulate MBNL protein levels and that targeting miRNAs is a potential treatment for DM. However, I have the following concerns.

Major Concerns:

1. Fig. 4.
a. Fig. 4A. Western blots of non-transfected DM1 and unaffected cells treated with scrambled

oligos should be included as additional controls.

b. Fig. 4G-J. This figure clearly shows the MBNL1 localized to the nucleoplasm in control myoblasts and in nuclear RNA foci in DM1 cells but after antagomiR-23b or -218 treatment the protein outside of the remaining foci appears to be distributed throughout the cells and is not concentrated in the nucleus. Do the authors have an explanation for this observation?

2. Fig. 5b-e. The statement that antagomiRs show a strong punctate pattern in nuclei should be confirmed by DAPI staining and the authors also need to provide evidence that these are myonuclei?

3. Fig. 6. It is important to target a control miR to confirm that splicing changes do not result from non-specific effects of systemic antagomiR administration.

4. Discussion. The authors need to address why their results vary from a previous 2010 paper (Genes Dev 24:653), which is not cited in this manuscript, that identified miR-23b in mouse heart as a developmentally regulated miRNA that directly regulates CELF1 and CELF2 levels but not MBNL1. Does miR-23b have different targets in the heart and skeletal muscle? If antagonizing miR-23b in the heart upregulates CELF1 and CELF2 how should this be addressed in a potential therapeutic treatment? It would be more convincing that CELF1 is not affected in the current study if some of the same splicing events were checked as were checked by the prior publication (e.g., Sorbs1, Mfn2, Mtmr3, H2afy, Ank2). In addition, Ref 10 (Kalsotra et al) reported that miR-23 is downregulated in the heart with CTG repeat expression. In the current study, changes in miR-23 levels in unaffected vs disease control vs DM1 are not shown and how repeat expression mediated downregulation of miR-23 would affect treatment potential of miR-23 antagonists is not discussed.

Minor Concerns:

1. Fig. S2. Nuclear and cytoplasmic markers should be included.

2. M&M. p12. The authors state that both PM and MUT sequences are included in Table S2 but this is incorrect.

3. Figs. 3, 4. It would be informative to state how many CTG repeats are present in these DM1 cells.

Response to Reviewers' comments:

We want to thank the reviewers for their comments and constructive criticism. In order to address them we have performed additional experiments, revised the text, and included their suggestions, all of which has helped to make a much stronger manuscript. We note the inclusion of a new author as a result of all changes that have been made.

Reviewers' comments:

Reviewer #1 (Remarks to the Author):

This paper describes the identification of microRNAs and antagomirs to target them to affect expression of MBL1 protein in DM1. The authors have used model cellular systems to identify a series of microRNAs that regulate the expression of the protein. These targets were first validated, again, in a model cellular system. Then, the targets were measured for their expression in DM1 patient cells and also in the mouse model for DM1. The application of the antagomirs boost expression of MBNL1 protein. In addition, these studies were applied into animal models of disease. The experiments were carefully carried out.

a few comments:

1. The citations of the small molecule approaches that target CUG repeats is incomplete and much of the more advanced approaches have not been cited properly. For example, a recent Nature chemical biology paper for various approaches to target should be referenced. Additionally, there are several approaches such as transcriptional inhibition, HDAC inhibition, and also RNA targeting that should be better described to provide a landscape type of description of the approaches.

Following the suggestion of the reviewer we have added more information regarding other therapeutic approaches targeting CUG repeats in DM, and included more updated references in this part of the Introduction section:

“Numerous therapeutic approaches have been designed following different approaches^{18,19} that can be broadly grouped as; (1) specific targeting of the mutant allele or its RNA product, including preventing MBNL protein sequestration using small molecules²⁰⁻²³, transcriptional^{24,25} and post-transcriptional silencing of DMPK²⁶, and (2) target signaling pathways downstream from CUGexp expression^{27,28}. Two strategies have reached human trials: Tideglusib, a small molecule non-ATP-competitive glycogen synthase kinase 3 (GSK-3) inhibitor⁴ (clinical trial NCT02858908) and IONIS-DMPKRx, an RNase H1-active ASO that target CUGexp RNA²⁹ (clinical trial NCT02312011). However, in IONIS-DMPKRx clinical trial, drug levels measured in muscle biopsy confirmed that the amount of target engagement was not enough to exert³⁰.”

2. The targeted microRNAs are involved in a myriad of processes. For example, 23b is a tumor suppressor and 218 is deleted in various cancers. Thus, what is the effect long term on other proteins in these microRNA-mediated pathways and also what are the effects on these proteins. This is an important consideration that is not even described in the manuscript.

Given the increased stability of phosphorotioate modified oligos (Nucleic Acids Res. 1995 Oct 25; 23(20): 4134–4142) we agree with the reviewer on his/her concerns regarding the long-term effects of this therapy. In order to assess the long-term effects of the inhibition of these microRNAs, we included in the revised version a new group of treated mice, which were sacrificed 6 weeks after the injection of the antagomiR molecules. In these animals we quantified MBNL expression, and microRNA levels in muscle to check the duration of the antagomiR effects at the molecular level, and myotonia and force to assess the functional relevance of these long-term effects. All these new data have been included in new Fig 7 and a new Results section.

3. What are the overall effects of over expression of MBNL1? There is also a loss of DMPK and more MBNL1 could cause more of the target repeat to be retained in the nucleus. Does that mean more of the translational defect of this protein is observed, does it also mean that the RNA is more abundant because loaded with more protein. This should be described and these effects explained in the text.

To assess the effects of MBNL overexpression on mutant RNA abundance, we quantified foci number and the percentage of cells with foci in DM1 cells treated with antagomiRs and observed that they were not significantly different from the values obtained in untreated cells. These data have been added to new Fig 4 and the corresponding Results section:

“Because the relationship between MBNL proteins and CUGexp foci formation is complex a potential undesirable side effect of boosting MBNL expression was an increase in the number of ribonuclear foci. To specifically test this hypothesis, we quantified foci in antagomiR treated DM1 fibroblasts and found that remained unaltered (Fig. 4 o-r).”

4. Oligonucleotides-based approaches have been put into the clinical for DM1 and they have been taken out of the clinic because of a lack of muscle penetration into human muscle. This should be described and an explanation of why this approach could be better, or why likely it is not, should be vetted out in the paper.
- 5.

A summary of the reports of these clinical trial has been included in the Introduction Section:

“... IONIS-DMPKRx, an RNase H1-active ASO that target CUGexp RNA²⁹ (clinical trial NCT02312011). However, in IONIS-DMPKRx clinical trial, drug levels measured in muscle biopsy confirmed that the amount of target engagement was not enough to exert³⁰.”

6. Other microRNAs could be targeted with the oligonucleotides at the dosage given and, yet, there are no selectivity studies done. For example, microRNAs that have overlapping seed sequence or are partially complimentary should be studied and their effects measured. This is important for a variety of reasons, and this is an additional consideration to point 2 above.

To assess the specificity of the antagomiR treatment, in the revised version, we quantified the amount of *miR-218* and *miR-23a* in cells and mouse muscles treated with antagomiR-23b. As expected, *miR-218* was not modified by the treatment but *miR-23a*, which has the same seed region with *miR-23b*, was significantly reduced. *miR-218* has not miRNAs with its same seed region. This data was included in Supplementary figure 4, and in the corresponding Results section:

“Importantly, levels of miR-218 were not altered in cells treated with antagomiR-23b, suggesting a specific effect of the antagomiR on its target. miR-23 family includes miR-23a and miR-23b, which are transcribed from different chromosomes, have identical seed

sequences, and differ by only 1 nucleotide on their 3' ends. As expected, antagomiR-23b also reduced the levels of miRNA-23a in the cells (Supplementary Fig 4 a-b)".

Reviewer #2 (Remarks to the Author):

Cerro-Herrero et al. shown that miR-23b and miR-218 repress MBNL1/2 and MBNL2. Using antagomiR against miR-23b and miR-218 there was an upregulation of the MBNL proteins which rescued the alternative splicing in DM1 human myoblasts. In vivo antagomir administration in the HSA mouse model upregulated Muscleblind-like protein in gastrocnemius and quadriceps muscles. This was associated with an attenuation of impaired molecular signaling and improvements in muscle function.

Questions

The control mice received an injection of PBS. A more appropriate control would be the injection of a non-specific scramble antagomir. What is the justification for injecting only PBS and not a control antagomir?

We used PBS because we considered that any scrambled oligonucleotide could potentially exert unspecific deleterious effects in the mice. However, in response to the reviewer concerns, in the revised version we included a group of mice injected with a scrambled control oligonucleotide at both short (4 days after injection), and long-term treatment (6 weeks after injection). All the parameters measured in this group were similar to the ones obtained in the mice injected with PBS except for total bilirubin and monocytes values in blood, which indicate a sequence-independent effect by the oligo chemistry. This data has been included in the Supplementary Fig 8.

Why would the antagomiR administration only ameliorate aberrant exon choices for Atp2a1 (exon 22) and Nfix (exon 7), and increased Clcn1 exon 7a PSI in gastrocnemius but not in quadriceps of HSALR mice. Does this suggest that such an approach would not be therapeutically relevant for humans?

In response to the next concern by the reviewer about the reduced number of animals in experimental groups, we repeated the experiments with 5 more animals of per group. The new data about splicing correction, including the new groups of animals, shows similar levels of rescue in gastrocnemius and quadriceps (Supplementary Fig 9). However, a situation where some splicing events are corrected in some tissues and not in others is definitely possible in human patients given the difference in the repeat expression levels and length resulting from the intrinsic mosaicism of the disease. In muscles with higher expression of repeats or expressing longer expansions, MBNL proteins available would be more reduced and the amounts of MBNL expression needed to rescue the pathologic effects would be higher than in other muscles.

Statistics

Why have the authors only used 4 mice per group? What is the justification for such a small sample size?

In response to the reviewer's concern, we repeated our experiments with new groups of antagomiR-treated and control mice. In the new Fig 5, 6, Supp Fig 9, Sup Fig 10 and Supp Fig 11 the data analyzed now includes 9 mice injected with antagomiR-23b, 9 mice injected with antagomiR-128 and 10 PBS-injected mice. The new sample size has been included in figure legends and material and methods section.

Page 8, line 3. "..., antagomiR-23b rescued Nfix and Clcn1 splicing to a greater extent than antagomiR-218 in quadriceps, which correlated with the lower upregulation of Mbnl1 and 2

protein levels achieved by antagoniR-218 in this muscle”. How reliable is a correlation with only 4 data points?

In the revised version, we show the correlations with the new sample size in Supp Fig 10, and the correlation is more reliable.

Figure 5a. Why was only n=3 analysed for the PCR? The figure legend states that 5 and 4 mice were used.

The animals used in Fig 5a were three FVB mice different from the ones used in the treated groups (Fig 5 f to 5 o), which were all DM1 model mice, 5 of them treated with PBS and 4 treated with each antagoniR. In the revised Fig 5, the n number has increased and now 10 animals have been treated with PBS and 9 animals have been treated with each antagoniR. This has also been clarified in the figure legend.

Discussion

Page 9, second paragraph. The authors state that no detrimental phenotype was observed over the 4 day experimental period. What exactly were the authors looking for that would suggest a “detrimental phenotype”? What happens if the mice are analysed several weeks after the antagoniRs are injected? This is important from a pre-clinical perspective.

In the previous group of short-term treated mice, our veterinary performed a visual necropsy and no tumours or tissue alterations were observed. In the new group of long-term treated mice, in addition to the necropsy, complete blood count with differential, and biochemical parameters measured in the blood of the mice before sacrifice. All this data has been included in a new Results section and Table S3.

Also, the authors state that “Targeting miRNAs in TGM is also advantageous because it regulates genes in specific tissues “. I fail to see how this approach is tissues specific. This becomes even more relevant as the authors show that the approach does not target all muscles. This suggests that this approach would not be suitable for humans. Please comment.

We agree with the reviewer that this sentence is misleading, we meant that as miRNAs expression is tissue-specific the gene modulation is only possible in the tissues where the specific miRNA is expressed. To avoid misunderstandings, we have replaced this sentence by:

“Given the tissue-specific expression of microRNAs, their targeting in TGM is also advantageous because it regulates genes only in the places where the regulator miRNAs are expressed, and the intensity of the upregulation will depend on the degree of repression by that microRNA.”

The reviewer concerns regarding the suitability of the treatment in humans has been already addressed in the reply to his/her comment 2.

Reviewer #3 (Remarks to the Author):

Following their earlier studies in *Drosophila*, Cerro-Herreros and colleagues screened for human miRNAs that modulate MBNL1 and MBNL2 levels in HeLa cells and identify miR-96, miR-23b and miR-218 as novel miRs that decrease MBNL RNA and protein levels. Luciferase reporter assays with MBNL 3' UTRs confirmed this regulatory activity and when predicted binding sites were mutated, or made perfect matches, regulation was altered accordingly. Although miR-23b and miR-218 were expressed in myoblasts and muscle, miR-96 was not and sampling of mouse tissues confirmed that miR-23b and miR-218 were expressed at relatively high levels in the majority of tissues examined so these miRs were pursued further. In the interest of developing treatments that increase MBNL protein levels, antagonists of candidate miRNAs were designed.

ASO/antagomiRs against -23b and -218 increased DM1 myoblast MBNL RNA and protein levels and rescued missplicing of some MBNL target RNAs. AntagomiR treatment of HSALR mice also increased MBNL protein levels in two skeletal muscles and correlated with improvements in myotonia and histopathology as well as partial rescue of alternative splicing. Overall, this is a novel and important study that demonstrates these miRNAs modulate MBNL protein levels and that targeting miRNAs is a potential treatment for DM. However, I have the following concerns.

Major Concerns:

1. Fig. 4.

a. Fig. 4A. Western blots of non-transfected DM1 and unaffected cells treated with scrambled oligos should be included as additional controls.

A new supplementary figure showing the western blots including the mock-transfected DM1 and unaffected cells (TR) and scrambled-transfected DM1 and unaffected cells (SC) has been added to the new revised version (Supplementary Fig 7).

b. Fig. 4G-J. This figure clearly shows the MBNL1 localized to the nucleoplasm in control myoblasts and in nuclear RNA foci in DM1 cells but after antagomiR-23b or -218 treatment the protein outside of the remaining foci appears to be distributed throughout the cells and is not concentrated in the nucleus. Do the authors have an explanation for this observation?

We have previously shown in our *Drosophila* model (Cerro et al. 2016) that microRNAs regulating *Drosophila* Muscleblind are isoform-specific. These data suggest that also in human cells, microRNA inhibition could have a differential influence on specific isoforms, which could originate for example, higher amounts of MBNL isoforms that localize in cytoplasm in the case of the treated cells.

2. Fig. 5b-e. The statement that antagomiRs show a strong punctate pattern in nuclei should be confirmed by DAPI staining and the authors also need to provide evidence that these are myonuclei?

Following the reviewer's suggestions, we have added the DAPI staining in these images.

3. Fig. 6. It is important to target a control miR to confirm that splicing changes do not result from non-specific effects of systemic antagomiR administration.

We have addressed a similar concern by Reviewer 2. We have included experiments with a scrambled control oligo in the revised version in order to discard non-specific effects.

4. Discussion. The authors need to address why their results vary from a previous 2010 paper (Genes Dev 24:653), which is not cited in this manuscript, that identified miR-23b in mouse heart as a developmentally regulated miRNA that directly regulates CELF1 and CELF2 levels but not MBNL1. Does miR-23b have different targets in the heart and skeletal muscle? If antagonizing miR-23b in the heart upregulates CELF1 and CELF2 how should this be addressed in a potential therapeutic treatment?

Not only miRNA levels but also the levels of the microRNA targets can be different in different tissues, as tissue-specific regulators may intervene. In addition, the variable repeat lengths from one tissue to another, leading to a very variable symptomatology in humans difficults the prediction of the therapeutic potential of this approach. We now included a short discussion on paper mentioned by the reviewer:

“Importantly, MBNL protein overexpression had previously shown to be well tolerated in mice models³⁴. Accordingly, in our study, upregulation of MBNL proteins through miR-23b or miR-218 silencing was not harmful, as mice showed no detrimental phenotype six weeks after the treatment. Similarly, miR-23b silencing in the heart of an inducible DM1 mice model caused no overt phenotype, despite producing CELF1 overexpression⁵⁹. This data supports the safety of our therapeutic approach in DM. Although different effects of antagomiR-23b on CELF1 levels are reported in these two studies, antagomiR dose used or tissue-specific effects might explain this controversy. Given the tissue-specific expression of microRNAs, their targeting in TGM is also advantageous because it regulates genes only in the places where the regulator miRNAs are expressed, and the intensity of the upregulation will depend on the degree of repression by that microRNA.”.

It would be more convincing that CELF1 is not affected in the current study if some of the same splicing events were checked as were checked by the prior publication (e.g., Sorbs1, Mfn2, Mtmr3, H2afy, Ank2). In addition, Ref 10 (Kalsotra et al) reported that miR-23 is downregulated in the heart with CTG repeat expression. In the current study, changes in miR-23 levels in unaffected vs disease control vs DM1 are not shown and how repeat expression mediated downregulation of miR-23 would affect treatment potential of miR-23 antagonists is not discussed.

We are already working on defining the role of miR-23 and miR-218 in DM pathology, which will be the focus of a different manuscript. For this reason, we preferred not to include this data on the present work. Here, we clearly showed in muscle that the levels of miR-23b inhibition that we achieve are not affecting the levels of CELF1, and we have included the quantification of two additional CELF1-dependent splicing events to strengthen our conclusion. In the study by Kalsotra *et al.*, the dose of antagomiR-23b used was 2-fold ours and this could be one potential explanation of the difference observed.

Minor Concerns:

1. Fig. S2. Nuclear and cytoplasmic markers should be included.

We have included DAPI as nuclei marker and now the signal was clearly observed in both compartments.

2. M&M. p12. The authors state that both PM and MUT sequences are included in Table S2 but this is incorrect.

The reviewer was right, the sequences were missing. We have included these sequences in new supplementary info 1 (SI1).

3. Figs. 3, 4. It would be informative to state how many CTG repeats are present in these DM1 cells.

We have included this information in the Material and method section.

REVIEWERS' COMMENTS:

Reviewer #1 (Remarks to the Author):

The paper is fine now. It is s solid but not a spectacular paper.

Reviewer #2 (Remarks to the Author):

No further comments.

Reviewer #3 (Remarks to the Author):

My concerns noted in the prior review have been adequately addressed and the revised manuscript represents a significant improvement.